# COMPENSATING FOR NONLINEAR REDUCTION WITH LINEAR COMPUTATIONS IN PRIVATE INFERENCE

## ABSTRACT

Increasingly serious data privacy concerns and strict regulations have recently posed significant challenges to machine learning, a field that hinges on high-performance processing of massive user data. Consequently, privacy-preserving machine learning (PPML) has emerged to securely execute machine learning tasks without violating privacy. Unfortunately, the computational cost to securely execute nonlinear computations in PPML models remains significant, calling for new neural architecture designs with fewer nonlinear operations. We propose Seesaw, a novel neural architecture search method tailored for PPML. Seesaw exploits a previously unexplored opportunity to leverage more linear computations and nonlinear result reuse, in order to compensate for the accuracy loss due to nonlinear reduction. It also incorporates specifically designed pruning and search strategies to efficiently handle the much larger design space including both nonlinear and linear operators. Compared to the previous state-of-the-art PPML for image classification on ImageNet, Seesaw achieves $1.68\times$ less latency at 71% iso-accuracy, or 4.59% higher accuracy at iso-latency of 1000K ReLU operations.

## 1 INTRODUCTION

Machine learning (ML) has become an indispensable and ubiquitous technology in contemporary data-driven applications, with deep neural networks achieving remarkable success in complex tasks such as image/video classification and natural language processing LeCun et al. (2015). The effectiveness of ML hinges on massive training data and extensive computational resources to efficiently process large network models. Consequently, ML tasks start to be outsourced to remote servers and deployed on cloud computing systems Aliyun; Amazon Web Services; Azure; Baidu; Cloud; OpenAI (2023). While cloud-based ML services bring a new revolution, this deployment model has also raised serious concerns regarding the privacy of user data, such as health/medical records, financial status, and location information, which must now be sent to public cloud platforms and suffer from leakage risks.

In response to the growing privacy concerns associated with ML applications, privacy-preserving machine learning (PPML) solutions have been proposed to securely store and process users' sensitive data without compromising confidentiality and integrity. State-of-the-art PPML frameworks heavily use cryptographic primitives, including homomorphic encryption and multi-party computation, to achieve provable security Badawi et al. (2018); Brutzkus et al. (2019); Chandran et al. (2022); Dowlin et al. (2016); Juvekar et al. (2018); Liu et al. (2017); Mishra et al. (2020); Ng & Chow (2021); Riazi et al. (2018); Zhang et al. (2023). However, despite extensive algorithm and system optimizations, their computational cost is still several orders of magnitude higher than the original plaintext models, resulting in unacceptably long execution latency that restricts their practical usage in time-sensitive scenarios like online inference. The high processing overheads are primarily associated with nonlinear operators (*e.g.*, activation functions such as Sigmoid and ReLU), which involve complex secure multi-party computation protocols Yao (1986) with heavy cryptographic computations (*e.g.*, AES encryption) and frequent communication between the user and the cloud.

Great efforts have thus far been made to alleviate the nonlinear computational cost in PPML, such as developing more efficient protocols for nonlinear operators Ghodsi et al. (2021); Lou et al. (2021); Mishra et al. (2020), or reducing the number of such operations through pruning and neural architecture search (NAS) Cho et al. (2022a;b); Ghodsi et al. (2020); Huang et al. (2022); Jha et al. (2021); Kundu et al. (2023a;b). Nevertheless, almost all prior techniques simply started with an

existing network architecture, and only focused on *reducing* the amount of nonlinear operators while struggling to minimize the corresponding negative accuracy impact. This approach inevitably causes increasing accuracy degradation when more nonlinear computations are reduced, suffering from the fundamental tradeoff between model accuracy and execution latency.

**Our contributions.** In this work, we aim to break this tradeoff, by exploiting opportunities to *use additional computations and data orchestration to compensate for accuracy loss due to nonlinear reduction*. Specifically, we propose two approaches: (1) *adding more linear operations* to the model to recover its decreased representation capacity; (2) *reusing the results of the remaining nonlinear operators* as much as possible through introducing residual shortcut links to the model topology. Although adding such linear and aggregation computations would increase the execution latency in the insecure case, the overheads in the PPML scenario are negligible compared to the dominant nonlinear cost, therefore exhibiting a unique opportunity.

We thus design Seesaw, a one-shot NAS method that leverages the above compensation ideas to automatically search for optimized neural network architectures for PPML. Besides the existing problem of determining how to selectively enable nonlinear operations under a given nonlinear budget, Seesaw needs to further deal with several new challenges. First, it must decide the amounts of extra linear computations and data reuse to add, in order to achieve a balance between sufficient representation capacity and overfitting avoidance. We propose novel pruning and NAS techniques to solve this issue. Second, it also needs an efficient search and training strategy, because the overall design space is significantly enlarged with the additional computations. We present a novel search strategy with a modified loss function. When evaluated on the CIFAR100 and ImageNet datasets under a wide range of nonlinear budgets, Seesaw is able to push the Pareto optimal frontier between the model accuracy and the execution latency. Compared to the previous state-of-the-art Kundu et al. (2023a), Seesaw achieves $1.68\times$ latency reduction at iso-accuracy, or $4.59\%$ accuracy at iso-latency.

## 2 BACKGROUND

Privacy-preserving machine learning (PPML) aims to address the challenges of processing private user data on proprietary ML models, while not revealing any sensitive information to malicious participants during the computation. We focus on PPML inference. More specifically, privacy is protected if (1) the user learns no knowledge of the ML model except for the inference result of her own input data; and (2) the model owner gains no information about the user data.

Currently, there are mainly two approaches to realize PPML. Hardware-based trusted execution environments (TEEs) can protect sensitive data Hunt et al. (2018); Hynes et al. (2018); Kim et al. (2020); Kunkel et al. (2019); Li et al. (2021); Tramer & Boneh (2019), but TEEs are vulnerable to side channels, weakening their security Chen et al. (2019); Wang et al. (2018). Cryptography-based PPML protects data privacy using modern cryptographic primitives Damgård et al. (2012); Gentry (2009); Yao (1986). They offer theoretically provable, strong security guarantees. Our work optimizes the execution latency of crypto-based PPML solutions while minimizing the accuracy impact.

### 2.1 CRYPTOGRAPHIC PRIMITIVES AND PPML PROTOCOL

Existing PPML algorithms have used various cryptographic primitives to best match different computation patterns in ML applications. **Fully Homomorphic Encryption (FHE)** Gentry (2009) is a technique that allows for applying arbitrary functions composed of addition and multiplication on encrypted data (*e.g.*, user data or model weights). FHE is useful in PPML as linear operators (matrix multiplications, convolutions, *etc*.) account for a majority of computations in modern ML models. Previously, CryptoNets Dowlin et al. (2016), HCNN Badawi et al. (2018), TAPAS Sanyal et al. (2018), LoLa Brutzkus et al. (2019), and Faster CryptoNets Chou et al. (2018) have explored the application of FHE in PPML. Unfortunately, the computational complexity of FHE is quite high and can result in several orders of magnitude slowdown compared to insecure computing.

Another way to support linear computations is **Secret Sharing (SS)** Damgård et al. (2012). PPML typically assumes two parties, the user and the model owner. SS transforms the data of each party into randomly split *shares*. Each share is hold by one party, and each party only sees its own share but not the full value, ensuring data privacy. Addition of two encrypted values, as well as multiplication between an encrypted value and a plaintext number, can be done locally with only simple operations.

Therefore, the linear operators that involve the encrypted user data and the plaintext weights can be done efficiently. Gazelle Juvekar et al. (2018) and DELPHI Mishra et al. (2020) have used SS to replace FHE for higher online processing speed. Nevertheless, FHE is still needed during offline pre-processing to prepare the share values.

The remaining challenge is handling nonlinear operators such as ReLU and MaxPool. **Garbled Circuit (GC)** Yao (1986) takes the encrypted boolean representations of the two parties' input data, and securely computes an arbitrary boolean function composed of AND and NOR gates. Most existing PPML systems use GC to compute nonlinear operators Juvekar et al. (2018); Liu et al. (2017); Mishra et al. (2020); Mohassel & Zhang (2017); Rouhani et al. (2018). GC processing requires heavy cryptographic computations (*e.g.*, AES encryption) and frequent communication between the two parties, and thus incurs significant overheads compared to insecure nonlinear processing.

**PPML protocol.** In this work, we follow the overall execution flow of the state-of-the-art PPML system, DELPHI Mishra et al. (2020). The protocol consists of two phases: an offline pre-processing phase, and an online inference phase. During offline pre-processing, we use FHE algorithms to generate the secret shares that will be used by the online SS scheme to compute the linear operators. Specifically, for a linear operator $\mathbf{y}_i = \mathbf{W}_i \cdot \mathbf{x}_i$, the user and the model owner each randomly samples a vector, $\mathbf{r}_i$ and $\mathbf{s}_i$, respectively. The user sends $\mathsf{Enc}(\mathbf{r}_i)$ (encryption of $\mathbf{r}_i$) to the model owner, who homomorphically computes $\mathsf{Enc}(\mathbf{W}_i \cdot \mathbf{r}_i - \mathbf{s}_i)$ using FHE. The user receives and decrypts this result to keep $\mathbf{W}_i \cdot \mathbf{r}_i - \mathbf{s}_i$. We also generate the GC boolean function for the nonlinear operators. For example, the user creates a GC function $f(\mathbf{a}) = \mathrm{ReLU}(\mathbf{a} + (\mathbf{W}_i \cdot \mathbf{r}_i - \mathbf{s}_i)) - \mathbf{r}_{i+1}$ for the ReLU operator $\mathbf{x_{i+1}} = \mathrm{ReLU}(\mathbf{y}_i)$, and sends it to the model owner.

In the online inference phase of a linear operator, the two parties start with each holding a share of the input, *i.e.*, $\mathbf{r}_i$ by the user and $\mathbf{x}_i - \mathbf{r}_i$ by the model owner. These shares are either from the results of the previous operator, or the user calculates $\mathbf{x}_i - \mathbf{r}_i$ and provides it to the model owner if this is the first layer. The model owner then evaluates $\mathbf{W}_i \cdot (\mathbf{x}_i - \mathbf{r}_i) + \mathbf{s}_i$ on its share. The user already has $\mathbf{W}_i \cdot \mathbf{r}_i - \mathbf{s}_i$ from the pre-processing phase. We can verify that these two values are exactly the shares of the output, *i.e.*, summed up to $\mathbf{W}_i \cdot \mathbf{x}_i = \mathbf{y}_i$. Thus we have maintain the induction condition.

For nonlinear operators, the online inference uses GC. We use ReLU as an example, $\mathbf{x_{i+1}} = \mathrm{ReLU}(\mathbf{y}_i)$. The model owner has the GC function $f(\mathbf{a})$ from the offline phase. It sets $\mathbf{a}$ to its share of $\mathbf{y}_i$, *i.e.*, $\mathbf{a} = \mathbf{W}_i \cdot (\mathbf{x}_i - \mathbf{r}_i) + \mathbf{s}_i$, and then evaluates $f(\mathbf{a})$ (involving heavy computation and communication) to obtain $\mathrm{ReLU}(\mathbf{y}_i) - \mathbf{r}_{i+1} = \mathbf{x_{i+1}} - \mathbf{r}_{i+1}$, which is a valid share of the input to the next operator. The user holds the other share $\mathbf{r}_{i+1}$.

## 2.2 RELATED WORK

In the above PPML protocol, SS has made the online computations of linear operators almost as cheap as the original insecure processing, and GC offers general compute capability to support unmodified nonlinear operators to ensure the same accuracy level. However, the use of GC causes severe communication overheads, which become the main performance bottleneck (over $300\times$ slower than linear computations in DELPHI Garimella et al. (2022)). It is therefore necessary to focus on reducing the cost of nonlinear operators to speed up the PPML processing.

Recently there have been various proposals to address this issue. Some designs change the nonlinear operator computations from ReLU to more crypto-friendly alternatives. DELPHI Mishra et al. (2020) replaced part of the nonlinear operators with linear approximation to exploit the latency-accuracy tradeoff, using neural architecture search (NAS) techniques. SAFENet Lou et al. (2021) also used NAS to apply approximation, but at more fine granularity to reduce the accuracy impact. Circa Ghodsi et al. (2021) reconstructed ReLU into a sign test (by GC) plus a multiplication (by SS), in order to reduce the processing cost. Other solutions reduce (*i.e.*, prune) the amount of ReLU operators in existing neural network structures. CryptoNAS Ghodsi et al. (2020) rearranged the ReLU operators and used a macro-search algorithm, ENAS, to search for a network with fewer nonlinear operators. Sphynx Cho et al. (2022a) instead used micro-search approaches to design its building blocks more thoroughly to achieve higher accuracy. DeepReDuce Jha et al. (2021) pruned the model in a more fine-grained manner at the channel level, and further improved accuracy through knowledge distillation. SNL Cho et al. (2022b) was inspired by the parameterized ReLU and realized pixel-level ReLU pruning. SENet Kundu et al. (2023a) proposed the concept of ReLU sensitivity, which distinguished the importance of different nonlinear operators and realized automated ReLU pruning.

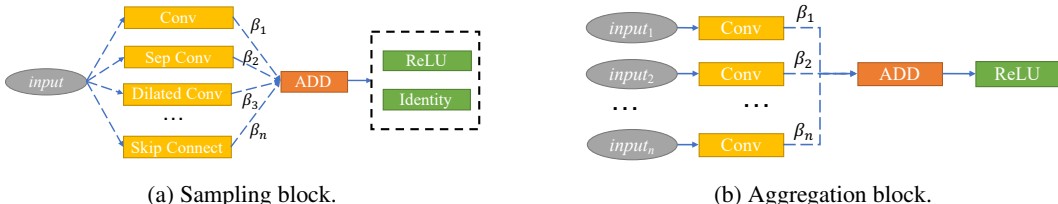

(a) Sampling block.  (b) Aggregation block.

Figure 1: Main building blocks of the Seesaw search space. The nonlinear ReLU operator is always placed after an element-wise ADD to save nonlinear computations.

## 3 DESIGN

Previous PPML designs that aimed to reduce the nonlinear cost (Section 2.2) suffered from a common limitation: they merely reduced the ReLU operators without reconsidering the overall network architecture. This inevitably decreases the representation capacity of the model. Since the representation capacity is jointly determined by both the linear and nonlinear operators, our key idea is to *compensate* for the accuracy loss caused by reduced nonlinear operators, by (1) *adding more linear operators in the model*, and (2) *reusing the remaining nonlinear outputs as much as possible*. We thus propose Seesaw, a one-shot NAS method to automatically search for crypto-friendly model architectures for PPML, with the best accuracy under the given budget for nonlinear operators, *i.e.*, the ReLU budget.

### 3.1 DESIGN SPACE

Seesaw uses two ways to compensate for the loss of nonlinear operators. Accordingly, two building blocks are added to its search space, as illustrated in Figure 1.

Figure 1a shows a **sampling block**, which substitutes a traditional Conv-ReLU block by enabling multiple parallel branches with various linear operators Szegedy et al. (2015; 2016). The branches can be convolutions with different kernel sizes (*e.g.*, $1 \times 1$, $3 \times 3$, $5 \times 5$), depth-wise separable convolutions, dilated convolutions, pooling, or even a direct skip connection. These independent branches enhance the model representation capacity by extracting multiple and different scales of features. While Sphynx Cho et al. (2022a) and CryptoNAS Ghodsi et al. (2020) also used up to four linear operators in a block, our sampling block is designed to contain much more branches to increase the expressivity. Note that all branches keep the original data shape and size, so their outputs can be weighted and combined with an element-wise ADD. The final ReLU may be pruned, *i.e.*, replaced with an Identify operator, to meet the overall ReLU budget, as described in Section 3.3.

Figure 1b shows an **aggregation block**, which aggregates the outputs of previous ReLU operators in the model. The goal of such aggregation is to maximally reuse the limited ReLU outputs remained in the pruned model, not only by the immediately succeeding block, but also potentially by all the following blocks, as shown in the overall supermodel in Figure 2. This helps prevent feature loss and overfitting Szegedy et al. (2015; 2016). Aggregating the ReLU outputs at different positions of the neural network like this is also another way to introduce extra nonlinear nature to the model. Each of these previous ReLU outputs first passes a convolution kernel to reduce the resolution. Then an element-wise ADD operator aggregates these data before feeding them to the final ReLU activation.

We point out two key points in both building blocks. First, both blocks place the (possible) nonlinear ReLU *after an ADD operator*. In contrast to the CONCAT operators used in CryptoNAS Ghodsi et al. (2020) and Sphynx Cho et al. (2022a), ADD results in a smaller data size after aggregation, and thus reduces the amounts of nonlinear operations for the following ReLU. Actually, because Seesaw intentionally employs a large number of branches, using CONCAT would lead to significantly higher cost for each ReLU (by a factor equal to the branch count), and thus limit the total number of ReLU operators allowed in the model. We present a detailed comparison in Section 4.2 to demonstrate the benefit. Second, in both blocks, the branches are accumulated according to certain *learnable weight parameters* $\beta_{i,j}$. We incorporate the training of these weights into the overall training process rather than separately determining them afterwards, as discussed later in Section 3.3. The weighted output also helps stabilize the training process, by suppressing the gradient explosion or vanishing issues.

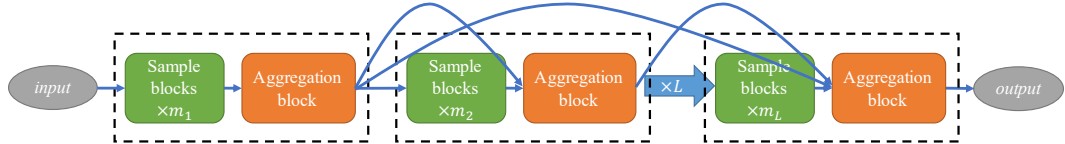

Figure 2: Supermodel architecture of Seesaw. There are $L$ units in total. Each unit contains $m_i$ sampling blocks followed by an aggregation block. The units are densely connected.

Finally, the sampling blocks and the aggregation blocks are used to construct an over-parameterized **supermodel** in Seesaw (Figure 2). Each aggregation block is preceded by several sampling blocks (*i.e.*, $m_i$). The output of each ReLU is forwarded to all the aggregation blocks after it through residual connections He et al. (2016), ensuring maximal nonlinear reuse. The use of residual connections not only avoids information loss and enables nonlinear operator reuse, but also speeds up training by preventing vanishing gradients. Several prior designs like CryptoNAS Ghodsi et al. (2020) and Sphynx Cho et al. (2022a) also used residual connections and mainly followed existing topologies like ResNet He et al. (2016) and NASNet Zoph et al. (2018). We emphasize that Seesaw uses much more residual connections beyond the original insecure network model, and for a completely different purpose of reusing the ReLU outputs in order to increase the representation capacity.

### 3.2 PRUNING METHODS

From the design space in Section 3.1, we see that the supermodel contains the following parameters: (1) The weight $\beta_{i,j}$ of the linear operator on the $j$-th branch in the $i$-th sampling/aggregation block. (2) The weight $\alpha_i$ (binarized to $\{0, 1\}$) to decide the nonlinear operator (ReLU or Identity) in the $i$-th sampling block. Seesaw applies pruning to the $\beta_{i,j}$ parameters of sampling blocks (but not aggregation blocks, see Section 4.3) and the $\alpha_i$ parameters of sampling blocks.

**Pruning linear branches.** Generally, we would need to prune the branches in each block to derive the final network architecture from the over-parameterized model. The pruning approach in traditional NAS in the insecure scenario is conservative, usually keeping only one of the multiple branches in each block, mainly to restrict the model size and the computation demand Cai et al. (2019); Liu et al. (2018); Wu et al. (2019). However, in PPML, the computational bottleneck does not lie in the linear operators. Therefore Seesaw could retain more branches in each block without worrying about the latency issue, thus increasing its representation capacity to compensate for accuracy loss.

On the other hand, pruning unimportant branches can help prevent model overfitting and improve its generalization ability. More linear operators do not guarantee improved accuracy. This is still an important issue in the PPML scenario. Therefore, Seesaw applies pruning to the branches in each block. Specifically, during training, Seesaw adopts a sparsity constraint to force the branch weights $\beta_{i,j}$ in the same block to become sparse. However, we cannot directly use the typical L1/L2 regularization which encourages all weights to be small. As discussed above, we still want to keep many important branches to improve the representation capacity, while only discarding unimportant branches. Therefore, we prefer some weights to be large while the others being small, *i.e.*, a distribution with large variance. So we propose a new penalty function $\mathcal{L}_{\text{lin}}$ to maximize the variance of the branch weights in each block,

$$\mathcal{L}_{\text{lin}} = -\sum_i \sigma^2[\beta_{i,j}, \forall j], \tag{1}$$

where $\sigma^2$ computes the variance. After finishing training, we prune the branches with weights smaller than an empirically determined threshold, *i.e.*, 0.001 in our experiments. Section 4.3 reveals the relationship between the pruning threshold and the model accuracy.

**Pruning nonlinear operators.** We also need to prune the total number of nonlinear ReLU operators in the model, by selectively enabling a subset of the sampling blocks to use ReLU, while the others use Identity operators. This is controlled by the weight $\alpha_i$ for the $i$-th sample block. Similar to ProxylessNAS Cai et al. (2019), these parameters are binarized every epoch to ensure only one between ReLU and Identity is activated while searching. The total nonlinear count of the supermodel is calculated based on whether this weight is enabled and the size of the corresponding intermediate

data. This count is used to penalize the model if deviating from the given ReLU budget $\mathcal{B}_{\text{ref}}$,

$$\mathcal{L}_{\text{nonlin}} = \left| \frac{\sum_i \alpha_i H_i W_i C_i - \mathcal{B}_{\text{ref}}}{\mathcal{B}_{\text{ref}}} \right|, \tag{2}$$

where $H_i$, $W_i$ and $C_i$ are height, width and number of channels of the feature map at the $i$-th layer, respectively. After training, the ReLU operators are kept or pruned according to the binarized weights.

## 3.3 SEARCH STRATEGY

The loss function of Seesaw incorporates the linear and nonlinear pruning methods in Section 3.2,

$$\mathcal{L} = \mathcal{L}_{\text{CE}} + \lambda_{\text{lin}} \times \mathcal{L}_{\text{lin}} + \lambda_{\text{nonlin}} \times \mathcal{L}_{\text{nonlin}} \tag{3}$$

where $\lambda_{\text{lin}}$ and $\lambda_{\text{nonlin}}$ are weighting hyperparameters. $\mathcal{L}_{\text{lin}}$ and $\mathcal{L}_{\text{nonlin}}$ are from Equations (1) and (2). $\mathcal{L}_{\text{CE}}$ is the original cross-entropy loss. Given a sampled network $\mathsf{M}$ and a data-label pair $(X, Y)$, the cross entropy between the prediction $\mathsf{M}(X)$ and the ground-truth label $Y$ is $\mathcal{L}_{\text{CE}} = \text{CE}(\mathsf{M}(X), Y)$. Such a loss function allows us to balance the model accuracy and the ReLU budget by simultaneously optimizing the loss value and regularizing the linear and nonlinear costs.

For the network architecture search strategy, traditional NAS typically constructs an over-parameterized supermodel encompassing all building blocks and potential branches in the search space. This supermodel contains numerous architecture parameters (*e.g.*, for each branch and for each block) that must be first sampled to generate a specific network to train Cai et al. (2019); Liu et al. (2018). The search space from which network architectures are sampled is too large, making the training converge slowly. Some approaches try to directly train on the dataset, then optimize via a specific search algorithm, and finally do retraining Tan et al. (2019); Zoph et al. (2018). This process can still be computationally intensive and time-consuming.

Seesaw uses a novel search strategy, which only includes the existence of nonlinear operators (*i.e.*, $\alpha_i$) in the search space, and treats the branch weights for the large number of linear operators (*i.e.*, $\beta_{i,j}$) similarly to the model weights and to be updated during training without extra sampling. This greatly reduces the search space, accelerating the convergence when searching the best network architectures.

Algorithm 1 shows the pseudocode of the Seesaw search algorithm. Seesaw takes the input training dataset $D_T$, the validation dataset $D_V$, and the nonlinear budget $B_{\text{ref}}$. It trains the supermodel and searches the network architecture iteratively in a continuous loop until converged. In each iteration, it samples a network architecture from the search space (*i.e.*, sample $\alpha_i$ values at Line 10), and uses the training dataset to train the network weights as well as the branch weights $\beta_{i,j}$ in the sampled model (Lines 9 to 12). After a certain number of warm-up training epochs, it starts to train the architecture parameters, *i.e.*, the NAS modules (Lines 2 to 8). The NAS modules are sampled to determine $\alpha_i$, *i.e.*, the existence of each ReLU operator (Line 4). We use the overall loss $\mathcal{L}$ from Equation (3) to update the NAS modules (Lines 6 and 7). The network weight parameters are now

---

**Algorithm 1** Seesaw network architecture search

**Input:** training dataset $D_T$, validation dataset $D_V$, nonlinear budget $\mathcal{B}_{\text{ref}}$
**Output:** optimized network architectures
1: **while** not converged **do**
2:      **if** epoch > # warm-up epochs **then**
3:          sample $(X_V, Y_V)$ from $D_V$
4:          $\mathsf{M}$ = nas_modules.sample()
5:          $\mathcal{L}$ = CE($\mathsf{M}(X_V), Y_V$)
6:          $\mathcal{L}$+ = $\lambda_{\text{lin}} \times \mathcal{L}_{\text{lin}} + \lambda_{\text{nonlin}} \times \mathcal{L}_{\text{nonlin}}$
7:          update(nas_modules, $\mathcal{L}$)
8:      **end if**
9:      sample $(X_T, Y_T)$ from $D_T$
10:     $\mathsf{M}$ = nas_modules.sample()
11:     $\mathcal{L}_{\text{CE}}$ = CE($\mathsf{M}(X_T), Y_T$)
12:     update($\mathsf{M}, \mathcal{L}_{\text{CE}}$)
13: **end while**

---

frozen. The use of the validation set enhances the robustness of the architecture. After converged, the optimized network architectures can be derived based on the trained supermodel.

## 4 EVALUATION

We compare Seesaw with several previous PPML methods, including DELPHI Mishra et al. (2020), CryptoNAS Ghodsi et al. (2020), Sphynx Cho et al. (2022a), SNL Cho et al. (2022b), SENet Kundu

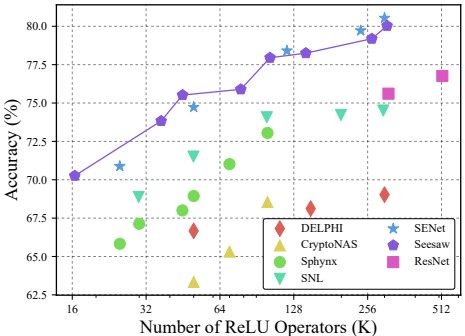

Figure 3: Overall comparison results of image classification on CIFAR100.

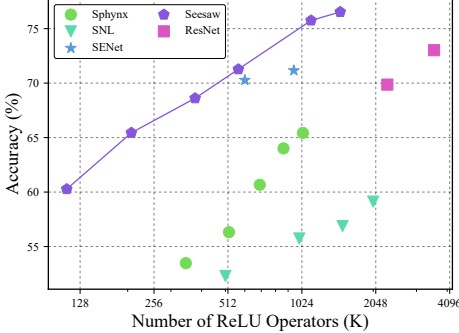

Figure 4: Overall comparison results of image classification on ImageNet.

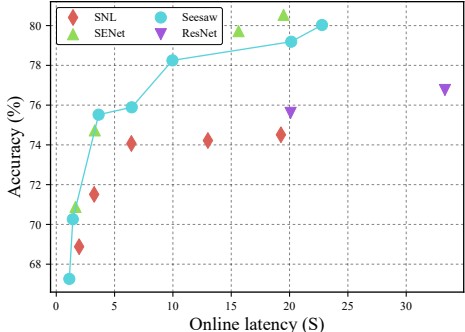

Figure 5: Real latency comparison results of image classification on CIFAR100.

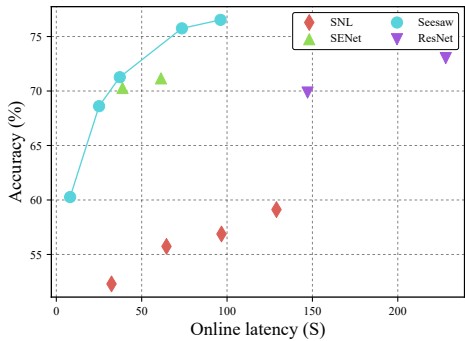

Figure 6: Real latency comparison results of image classification on ImageNet.

et al. (2023a), as well as unmodified baseline models. The baseline models are ResNet-18 and ResNet-34 He et al. (2016), with CIFAR100 Krizhevsky et al. (2009) and ImageNet Deng et al. (2009). We complete the search, training, and testing on machines with an Intel Xeon Gold 6145 CPU, 8 NVIDIA PH402 GPUs, and 1 Gbps Ethernet. We leverage the DELPHI framework to perform real performance experiments. We set 100 epochs for searching and 150 epochs for retraining with a decreasing learning rate from 0.05 to 0. $\lambda_{\text{lin}}$ and $\lambda_{\text{nonlin}}$ are initialized to 0.001 and 0.1, respectively.

### 4.1 COMPARISON WITH STATE-OF-THE-ART

Figures 3 and 4 show the comparison between our Seesaw and state-of-the-art PPML methods, on CIFAR100 and ImageNet, respectively. Following the common practice in previous work, we represent the runtime latency using the number of ReLU operators. The results clearly demonstrate the efficiency of Seesaw, in terms of the Pareto optimal frontier between the classification accuracy and the runtime latency. The ability to achieve higher accuracy with fewer nonlinear operators makes Seesaw a highly efficient and promising approach for PPML inference.

Specifically, on the CIFAR100 dataset (Figure 3), if we look at the accuracy level of $74\%$ for example, Seesaw only need about 36.8K ReLU operators, which are $1.36\times$ and $2.71\times$ fewer than the next best designs, SENet and SNL. On the other hand, when doing an iso-latency comparison at 50K ReLU, Seesaw improves the accuracy to $75.52\%$, which is $0.79\%$ better than SENet and $1.45\%$ better than SNL. The improvements over SENet are relatively small, and sometimes Seesaw has worse accuracy than SENet at high ReLU budgets. This is because SENet applies more fine-grained pixel-level ReLU pruning, which reduces the accuracy loss but requires more complex search and training methods.

On ImageNet, Seesaw outperforms the other proposals more significantly. At iso-accuracy of $71\%$, Seesaw saves $1.68\times$ ReLU counts over SENet. At iso-latency of 1000K ReLU counts, Seesaw achieves $75.75\%$ accuracy, $4.59\%$ higher than SENet.

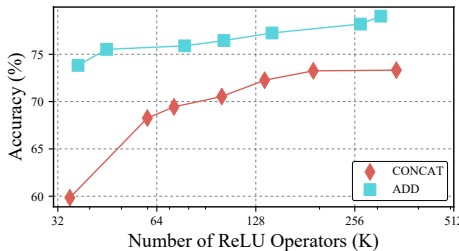

Figure 7: Comparison between the ADD and CONCAT operators on CIFAR100.

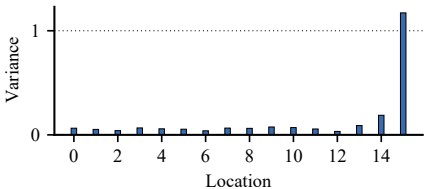

Figure 8: Branch weight variance distribution of sampling blocks at different locations on CIFAR100, under a ReLU budget of 36,684.

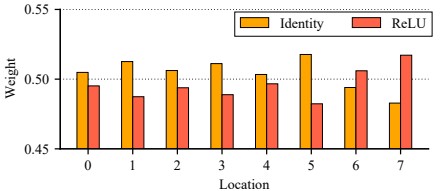

(a) Model-1, with a ReLU budget of 36,684.

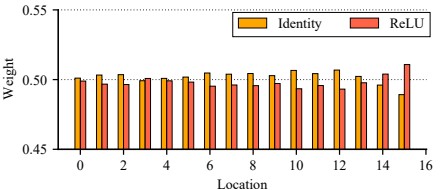

(b) Model-2, with a ReLU budget of 77,824.

Figure 9: Nonlinear operator weights of sampling blocks at different locations on CIFAR100. The one with a larger weight between ReLU and Identity will be used.

Note that when the ReLU budget is abundant, Seesaw can even outperform the accuracy of the original insecure ResNet models. This is expected because Seesaw uses more linear operators. In the insecure scenario, such accuracy increases come at the cost of longer inference latency. However in PPML, the latency is dominated by ReLU, for which Seesaw has similar or fewer operators.

Figures 5 and 6 do the above comparisons using real execution performance in terms of inference latencies. Even with the extra cost of computing more linear operators, Seesaw can still achieve better accuracy-latency tradeoffs compared to the baselines. The general trend in these figures is similar to the previous results using ReLU counts.

## 4.2 ABLATION STUDY: ADD VS. CONCAT

We compare the performance of ADD and CONCAT operators in Figure 7. We design another sampling block that is similar to Figure 1a but uses CONCAT instead of ADD. We then apply the same Seesaw search algorithm to find the best network architecture under different ReLU budgets and retrain the new models. For a fair comparison, we use the same ReLU budgets for the ADD- and CONCAT-based models. From the figure we see that, the CONCAT-based models can achieve good accuracies, but still not as high as the ADD-based models, exhibiting an average gap of 7.0%. The accuracy difference is particularly significant when the ReLU budget is tight. Essentially, using ADD operators allows for more linear operators in the model and thus higher expressivity *without* consuming extra nonlinear operators, which is more efficient.

## 4.3 ABLATION STUDY: PRUNING METHODS

Section 3.2 introduces how to prune linear operator branches in each sampling block. In our experiments, we initialize 27 branches of different linear operators in every sampling block. We evaluate three different pruning schemes: keeping all branches (**All**), keeping a fixed number of branches with the highest weights (**Fixed-1**, **Fixed-4**, **Fixed-11**), and keeping the branches whose weights exceed the threshold (**Threshold-0.1**, **Threshold-0.001**, **Threshold-0.00001**).

As shown in Table 1, **All** does not achieve the highest accuracy, while our **Threshold-0.001** method works the best. Removing branches with low contributions reduces the risk of overfitting. Comparing **All** and the several **Fixed** approaches, we see the effectiveness of using more linear operators for feature extraction to improve the model representation capacity and thus the accuracy. However,

Table 1: Model accuracy comparison of pruning methods for sampling blocks under different ReLU budgets on CIFAR100.

| # ReLU | Pruning method | Acc. (%) |
|---|---|---|
| | All | 72.63 |
| | Fixed-1 | 66.52 |
| | Fixed-4 | 72.02 |
| 36,684 | Fixed-11 | 72.53 |
| | Threshold-0.1 | 69.47 |
| | Threshold-0.001 | **73.83** |
| | Threshold-0.00001 | 73.06 |
| | All | 74.33 |
| | Fixed-1 | 71.21 |
| | Fixed-4 | 72.83 |
| 77,824 | Fixed-11 | 73.75 |
| | Threshold-0.1 | 70.41 |
| | Threshold-0.001 | **75.89** |
| | Threshold-0.00001 | 75.20 |

Table 2: Model accuracy comparison of nonlinear reuse methods under different ReLU budgets on CIFAR100.

| # ReLU | Reuse method | Acc. (%) |
|---|---|---|
| | None | 70.20 |
| 36,684 | Half | 72.64 |
| | All | **73.83** |
| | None | 72.53 |
| 77,824 | Half | 75.01 |
| | All | **75.89** |
| | None | 69.98 |
| 102,400 | Half | 75.41 |
| | All | **76.95** |
| | None | 72.33 |
| 143,360 | Half | 76.51 |
| | All | **77.25** |

**Fixed** cannot adapt itself to different sampling blocks at different locations. According to Figure 8, the branch weights of latter sampling blocks tend to have higher variances, which means fewer branches should be retained. Therefore, different sampling blocks prefer different numbers of linear operators, leading to the decision of using a threshold to prune the branches.

We further conduct an ablation study on the impact of nonlinear reuse residual links, *i.e.*, input paths of aggregation blocks. The results are listed in Table 2, where three methods are tested: using no nonlinear reuse (**None**), keeping half (50%) of the residual links with the highest weights (**Half**), and keeping all links (**All**). The results indicate that the **All** scheme achieves the highest accuracy under different ReLU budgets, while **None** exhibits different degrees of accuracy drop from 3.4% to 6.5%. As a result, different from sampling blocks that apply pruning, aggregation blocks in Seesaw keep all the reuse links activated. These results underscore the efficacy of nonlinear reuse in Seesaw.

## 4.4 NETWORK ARCHITECTURE ANALYSIS

Finally, we illustrate the distributions of the ReLU operators in the optimized network architectures discovered by Seesaw. Figure 9 shows the corresponding weight values for ReLU and Identity at different sampling blocks in two networks with different ReLU budgets. The sampling blocks at the latter stage of the network tend to have higher ReLU weights and would keep the ReLU operators. This observation aligns with the ReLU sensitivity observed in SENet Kundu et al. (2023a). For example, Model-1 in Figure 9a with a small ReLU budget only keeps the last two nonlinear operators. However, Seesaw can also retain some earlier nonlinear operators to if the ReLU budget allows, in order to boost the accuracy. For example, Model-2 in Figure 9b preserves the ReLU at location 3. In contrast, Figure 8 shows that the variance of sampling block branch weights is likely higher towards the backend of the network, reflecting that more linear operators are pruned under the threshold.

Combining the above two trends, we get to an interesting observation. *An optimized PPML network architecture needs to preserve sufficient nonlinearity in the latter blocks of the model, while at the earlier stage, it can instead increase the linear computations to increase the representation capacity. The two patterns compensate very well, once again validating the design principle of Seesaw.*

## 5 CONCLUSIONS

In this paper, we present Seesaw, a neural network structure search scheme that is tailored to private machine learning inference. Seesaw compensates for the negative accuracy impact of reducing expensive nonlinear operators through adding more linear computations and reusing existing nonlinear results. It incorporates novel pruning and search approaches to efficiently determine the optimized amounts of extra computation and data reuse. Our evaluation shows that Seesaw achieves higher accuracy with fewer nonlinear operations compared to previous proposals.

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
