# OpenReview forum: "Compensating for Nonlinear Reduction with Linear Computations in Private Inference"
_ICLR.cc/2024/Conference — Submitted to ICLR 2024_

### Official Review · Reviewer_SDAb · 2023-10-30

**Soundness:** 2 fair
**Presentation:** 2 fair
**Contribution:** 1 poor
**Rating:** 1
**Confidence:** 5

**Summary:**

This paper introduces approaches to improve the performance of cryptographically secure private inference by improving the network's ReLU efficiency. The authors leverage neural architecture search to compensate for the reduced number of ReLU activations by augmenting the network's linear operations (FLOPs). The achieved performance is on par with state-of-the-art ReLU-pruning methods.

**Strengths:**

$\bullet$  Experimental results on Imagnet dataset.

$\bullet$ The ReLU-Accuracy performance of the proposed baseline model is on par with existing ReLU-pruning methods.

$\bullet$  Ablation studies are presented well to show the efficacy of the pruning method and the merits of reusing ReLUs.


$\bullet$ Proposed methods are very well presented and easy to understand.

**Weaknesses:**

$\bullet$ **Novelty of the method:** The proposed approaches for improving ReLU efficiency at the expense of FLOPs count are not novel at all. Previous work, such as CryptoNAS and Sphynx, has already explored this concept by maintaining a constant ReLU count per layer, which in turn increases the FLOPs count (however, these studies did not provide detailed FLOPs count information). Furthermore, this paper fails to introduce any fresh insights or observations that shed light on how networks can trade ReLUs for FLOPs. The insights presented in Section 4.4 of the paper are already well-established.

$\bullet$ **FLOPs-cost are ignore:** This is the core-issue with the paper. The authors presented the results with online latency and ignored their impact on end-to-end latency, a trend also seen in works like SENet, SNL, Sphynx, DeepReDuce, and CryptoNAS. Nonetheless, these prior work on PI improved the ReLU efficiency of the given baseline networks (mostly ResNets and WideResNets) *without increasing their FLOPs counts* (Although WidResNets have 4x to 5x higher FLOPs than ResNets).

Delphi assumes that no matter how many FLOPs are there in networks, they can be processed offline. However, in real-world scenarios, private inference requests arrive at non-zero rates. Even at low arrival rates, processing the entire FLOPs offline becomes impractical due to limited computing resources, storage, communication bandwidth between server and client, and time constraints arising from the non-zero request-arrival rate. Consequently, offline costs are no longer truly offline, and FLOPs start affecting real-time performance, as illustrated in Figure 7 of the paper [1]. This effect can be exacerbated by networks with higher FLOP counts, as proposed by the authors.


**The argument of online vs. offline latency is only valid if the optimization is performed for a single private inference in isolation.** FLOP penalties can only be disregarded when there are no inference arrivals or when an accelerator offering more than 1000x speedup is employed. Even with complete FLOP parallelization, such as using LPHE in [1], end-to-end performance improves but does not eliminate FLOP costs.

The authors cite [1] to support the claim that ReLU is 300x costlier than FLOPs, but this argument is valid only for online overhead. When considering end-to-end latency, FLOPs are shown to be 4.8x more expensive than ReLUs, as demonstrated in Table 1 of [1]. Therefore, the authors should have provided a FLOPs comparison with ResNet18, WRN22x8, and CryptoNAS.

In summary, *this paper does not contribute any new perspectives on private inference and fails to advance the understanding of current gaps in the field, rendering it less relevant for the ICLR audience.*


[1] Garimella et al., "Characterizing and Optimizing End-to-End Systems for Private Inference," ASPLOS 2023.

**Questions:**

See the points in weakness.

Additionally, in line with the proposed method, increasing the network's width has been shown to  [2] [3] (however,  at the expense of FLOPs). How does the proposed method's trade-off between ReLUs and FLOPs counts compare to the straightforward approach of widening the baseline networks, such as ResNet18?



[2] Dollár et al., Fast and accurate model scaling. CVPR'21.


[3] Lee et al.,  Wide neural networks of any depth evolve as linear models under gradient descent. NeurIPS'19.

---

> ### Author Response · Authors · 2023-11-19
>
> **1. Novelty of the method. **
>
> Thanks for raising the concern. We agree that additional experiments to explore the tradeoff between FLOPs and ReLU counts can help provide more novel insights. However, we believe that our paper **already offers significant novelty and contributions**, and **the current experiments also shed some light on the above tradeoff**.
>
> For CryptoNAS and Sphynx, their approach deviates from the PPML setting we adopt, and reduces the amount of ReLU using NAS at the expense of accuracy loss. Although they can introduce more linear operators, they will suffer from more ReLU counts with **one-to-one building blocks** instead of **many-to-one building blocks**. In contrast, we employ additional linear computation and fully reuse nonlinear operators to compensate for the accuracy loss, and achieve even better performance.
>
>
> **2. FLOPs-cost are ignored.**
>
> Thank you for pointing out the ASPLOS’23 paper and the issue of offline HE computation cost for generating secret share triples. We do admit that in our design, more linear operators will incur more offline HE computations, and at high request arrival rates, it may be a bottleneck in the online throughput. However, the offline HE computations can be accelerated using layer-parallel HE (LPHE) by Garimella et al. or hardware accelerators.
>
> As shown in Figures 5 and 6, even with the extra cost of computing more linear operators, Seesaw can still achieve better accuracy-latency tradeoffs compared to the baselines on real execution performance.
>
>
> **3. The argument of online vs. offline latency is only valid if the optimization is performed for a single private inference in isolation**
>
> As exactly in the same ASPLOS’23 paper by Garimella et al., the offline latency can be greatly reduced by using more server computing resources to compute in parallel, even without expensive HE hardware accelerators.
>
> In the table below, we compare our Seesaw models with the Delphi ResNet-32-300K baseline (300K ReLUs), completely considering both online and offline cost. The Delphi offline phase uses the ASPLOS’23 LPHE approach, and needs 14.44 seconds.
>
> Because Seesaw proposes a spectrum of models with different tradeoffs, we choose two models: Seesaw-16K (16K ReLUs) that has **similar linear FLOPs (and thus similar offline HE cost) and similar accuracy** to the baseline, and Seesaw-45K that has **2x linear FLOPs and higher accuracy**. We assume the Seesaw models use the same amount of offline computing resources as the baseline above, so the offline HE time is proportional to the linear FLOPs.
>
> |                                 | Offline HE | Garbled Circuits (online) | Secret Sharing (online) | Communication (online) | Total |
> | ------------------------------- | ---------- | ---------------- | -------------- | ------------- | ----- |
> | Delphi (69% CIFAR-100 accuracy) | 14.44      | 24.87            | 0.65           | 5.33          | 45.29 |
> | Seesaw-16K (69% CIFAR-100 accuracy) | 16.12      | 0.96       | 0.73          | 0.84          | 18.65 |
> | Seesaw-45K (72% CIFAR-100 accuracy) | 26.33    | 2.01	|0.99	        |1.28	|30.61    |
>
> We see that even when limited to similar FLOPs as the baseline, Seesaw-16K can still save online ReLU cost while maintaining accuracy, due to a shallower network topology. It achieves **over 2.4x end-to-end (offline + online) speedup**.
>
> For Seesaw-45K, even the offline HE cost is 2x more, the online save is even more significant to compensate for it, enabling end-to-end speedups. Although the speedup here is not very high, it is actually a conservative result, because **we can easily accelerate the offline part by using more servers, while the online part is difficult to accelerate**.
>
>
> **4. How does the proposed method's trade-off between ReLUs and FLOPs counts compare to the straightforward approach of widening the baseline networks, such as ResNet18?**
>
> Thanks for your references. We may add such experiments for comparison in the final version. Wider networks can be helpful, as shown in SNL [1]. However, we also need to argue that widening the networks in Seesaw is not applicable because more channels will increase the consumption of ReLU budgets. We organize the parallel linear operators at some linear cost, but widening the networks results in both linear and nonlinear cost.
>
>
> [1]: Minsu Cho, Ameya Joshi, Brandon Reagen, Siddharth Garg, and Chinmay Hegde. Selective network linearization for efficient private inference. In the International Conference on Machine Learning, pp. 3947–3961. PMLR, 2022b.

---

> > ### Comment · Reviewer_SDAb · 2023-11-22
> > **Reply to the Authors' Rebuttal**
> >
> > Thank you for the rebuttal.
> >
> >
> >
> > The data presented in the rebuttal is cherry-picked (seesaw-16K and seesaw-45K). Those comparisons should be presented over a broad spectrum of  ReLU counts (till 150K to 200K on CIFAR-100). Most of my initial concerns are still not addressed, and I will keep my score.
> >
> > I hope the authors will include a detailed discussion and comparison of FLOPs with the previous methods, even when comparing only the online latency. Also, the argument of  **one-to-one building blocks vs. many-to-one building blocks** is unjustified; these are just different implementations to improve the ReLU efficiency at the expense of higher FLOPs count, not a point to show the novelty.

---

### Official Review · Reviewer_68ze · 2023-11-01

**Soundness:** 3 good
**Presentation:** 3 good
**Contribution:** 2 fair
**Rating:** 5
**Confidence:** 2

**Summary:**

This paper addresses the issue of accuracy loss due to approximating the activation function during the implementation of a privacy-preserving artificial intelligence model. It proposes a method for modifying the model to add convolution operation blocks or reuse activation results to restore accuracy. In the process of modifying this model, it suggests utilizing Neural Architecture Search to automate the model adjustments.

**Strengths:**

When implementing information security artificial intelligence models using homomorphic encryption or multi-party computation, the issue of accuracy loss due to activation function approximation has been extensively discussed in many papers and is one of the fundamental problems to address. The approach of solving this problem by designing a Neural Architecture Search (NAS) with meaningful exploration directions, such as the addition of linear operations or activation reuse, is considered a novel method. While papers using NAS algorithms have existed before, I believe there is a novelty in setting the exploration directions. I consider this a valuable method that can be effectively used in subsequent papers to design information security artificial intelligence models.

**Weaknesses:**

When reviewing the experimental results using this technology, I found it ambiguous whether the corresponding privacy-preserving machine learning model were actually implemented with the homomorphic encryption and the multiparty computation. If homomorphic encryption and multi-party computation were used, specific encryption parameters and communication amounts should be provided, but such information was not clearly presented. While the NAS algorithm itself is novel, the lack of thorough discussion on security during its implementation and validation makes this paper appear incomplete as a paper on privacy-preserving AI. Simply presenting a good algorithm may not be sufficient for approval if it's not clear whether actual cryptographic algorithms were used in the implementation.

**Questions:**

1) Please all of the details about the cryptographic parameters in your implementation.
2) Please give the communication costs for each result.
3) Did you implement your models with homomorphic encryption and multiparty computation? or did you only compute the expected runtime for your results without the implementation of the privacy-preserving machine learning system?

My rating will be changed with these questions.

---

> ### Author Response · Authors · 2023-11-19
>
> **1. Please list all the details about the cryptographic parameters in your implementation.**
>
> Thank you for pointing that out. We leverage the DELPHI framework to perform real performance experiments. We inherit its settings on our machines and get the real latency in Figure 5 and Figure 6, whose implementation works over the 41-bit prime finite field defined by the prime 2061584302081, and uses a 11-bit fixed-point representation (see https://github.com/mc2-project/delphi/tree/master).
>
>
>
> **2. Please give the communication costs for each result.**
>
> We break the communication cost into offline and online phases and show them as follows:
>
> | Model               | Offline Communication/s | Online Communication/s | Accuracy |
> | ------------------- | ----------------------- | ---------------------- | -------- |
> | DELPHI (ResNet-32)  | 73.42                   | 3.86                   | 69       |
> | model-1             | 4.01                    | 0.21                   | 67.26    |
> | model-2             | 9.02                    | 0.47                   | 72.5299  |
> | model-3             | 11.03                   | 0.58                   | 72.95    |
> | model-4             | 19.05                   | 1                      | 73.31    |
> | model-5             | 25.06                   | 1.32                   | 74.54    |
> | model-6             | 35.09                   | 1.85                   | 74.659   |
> | model-7             | 65.16                   | 3.43                   | 75.439   |
> | model-8             | 75.18                   | 3.96                   | 76.66    |
>
> Table: Communication latency breakdown of Seesaw on CIFAR100.
>
> | model | offline communication/s | online communication/s |
> |-------|------------------------|-------------------------|
> | model-1 | 26.18 | 1.45 |
> | model-2 | 87.25 | 4.85 |
> | model-3 | 130.88 | 7.27 |
> | model-4 | 258.85 | 14.38 |
> | model-5 | 340.29 | 18.90 |
>
> Table: Communication latency breakdown of Seesaw on ImageNet.
>
> From the tables, we see that the reduction of ReLU can save a lot of online and offline communication cost compared to DELPHI with 69% accuracy.
>
>
> **3. Did you implement your models with homomorphic encryption and multiparty computation? **
>
> We do not implement a new PPML system, but adopt the design of DELPHI (see https://github.com/mc2-project/delphi/tree/master), which is a state-of-the-art implementation of PPML using HE and MPC.

---

### Official Review · Reviewer_gfp9 · 2023-11-08

**Soundness:** 2 fair
**Presentation:** 3 good
**Contribution:** 3 good
**Rating:** 6
**Confidence:** 2

**Summary:**

The paper Seesaw presents a new approach in Privacy-preserving machine learning which overcomes state-of-the-art issues on the decrease of accuracy when reducing the amount of non linear operators for a given architecture. The authors designed a new Neural architecture search where they consider to use more linear computations and to reuse the results from non linear operators and thus compensate from having less of them. The introduced design also enables more representational capability of the model with the increase on the linear operators. The proposed design outperforms SOTA especially on Imagenet with better accuracy with fewer non linear operations.

**Strengths:**

- The paper is well presented with understandable figures on the design of Seesaw
- The proposed approach gives competitive results compared to SOTA. Especially on Imagenet, we observe a big improvement

**Weaknesses:**

- The authors do not provide an analysis on the weighting parameters used in the loss function regarding the pruning of the linear branches and non linear operators.
- The results on CIFA100 are less significant compared to the one on Imagenet
- It would have been nice to have an additional dataset for the evaluation to see the result tendency compared to CIFAR100 and Imagenet

**Questions:**

- Is there a reason which explains why the results on accuracies on Imagenet and CIFAR100 are different with respect to SENet? We observe better results of Seesaw compared to SENet on Imagenet
- Section 4.1, you say that with abundant ReLu budget, Seesaw can outperform Resnet models, do you prove this statement somewhere? or It is just the assumption knowing that you have more linear operators.

---

> ### Author Response · Authors · 2023-11-19
>
> **1. Analysis on the weighting parameters used in the loss function regarding the pruning of the linear branches and nonlinear operators.**
>
> Thank you for your question. We use Eq (1) to capture the loss from the variation of linear branch parameters, and Eq (2) to capture the nonlinear operator loss as the difference between the target ReLU budget and the current ReLU number. Combining these two losses, we introduce our optimization targets: finding the important linear operators and removing unnecessary nonlinear operators. We adjust $\lambda_{lin}$ and $\lambda_{nonlin}$ in Eq (3) to get the best network architecture (with dynamic learning rate from 0.05 to 0 for $\lambda_{lin}$ and 0.001 and 0.1 for $\lambda_{nonlin}$).
>
> Figure 8 and Figure 9 provide simple analysis on weight parameters. Figure 8 shows that the variance of sampling block branch weights is likely higher towards the backend of the network, reflecting that more linear operators are pruned under the threshold. Figure 9 shows the sampling blocks at the latter stage of the network tend to have higher ReLU weights and would keep the ReLU operators.
>
> Therefore, we get an observation that an optimized PPML network architecture needs to preserve sufficient nonlinearity in the latter blocks of the model, while at the earlier stage, it can instead increase the linear computations to increase the representation capacity.
>
>
> **2. The results on CIFAR100 are less significant compared to the one on Imagenet.**
>
> ImageNet requires more expressive networks for higher resolution pictures, and Seesaw provides more linear operators to achieve that. However, on CIFAR100 with low resolution, the benefits of adding linear operators are relatively low after obtaining good feature extraction capabilities. At this point, the ReLU budget is the real determinant of accuracy.
>
>
> **3. Is there a reason which explains why the results on accuracies on Imagenet and CIFAR100 are different with respect to SENet? We observe better results of Seesaw compared to SENet on Imagenet**
>
> The reason is similar to that of Question 2 above.
> SENet proposes fine-grained ReLU pruning algorithms, which can efficiently reduce the ReLU counts of existing networks. They choose ResNet18 and ResNet34 as targets, which already has enough expression ability for CIFAR100. However, linear operators of shallow ResNet are not enough for ImageNet.
> In summary, simple networks can achieve good results on small datasets, but not on large datasets.
>
>
> **4. Section 4.1, you say that with abundant ReLU budgets, Seesaw can outperform Resnet models, do you prove this statement somewhere? or It is just the assumption knowing that you have more linear operators.**
>
> We compare Seesaw with ResNet on Figure 3 and Figure 4, finding that with the same ReLU budget, Seesaw can outperform ResNet. However, we must admit that Seesaw achieves such performance with more linear operators.

---

> > ### Comment · Reviewer_gfp9 · 2023-11-22
> >
> > Thank you for your answers. You provided the details I was expecting.

---

### Meta-Review · Area_Chair_eHtZ · 2023-12-04

**Metareview:**

Summary: Authors propose to reduce the amount of non-linear operators (and increase more linear operators) for a given architecture without sacrificing the accuracy too much.

Strength: The proposed method outperforms SOTA on Imagenet with better accuracy with fewer non linear operations.

Weaknesses:
1. The data presented in the rebuttal is cherry-picked (seesaw-16K and seesaw-45K). Those comparisons should be presented over a broad spectrum of ReLU counts (till 150K to 200K on CIFAR-100).

2. The idea used in this paper does not seem novel. It would strengthen this aspect by introducing new insights that shed light on how networks can trade ReLUs for FLOPs.

3. It would be useful to provide a detailed discussion and comparison of FLOPs with the previous methods

**Justification For Why Not Higher Score:**

The reviewers' concerns seem sensible.

**Justification For Why Not Lower Score:**

NA

---

### Decision · Program_Chairs · 2024-01-16

Reject